# Development History, Structure, and Function of *ASR* (*Abscisic Acid-Stress-Ripening*) Transcription Factor

**DOI:** 10.3390/ijms251910283

**Published:** 2024-09-24

**Authors:** Yue Zhang, Mengfan Wang, Andery V. Kitashov, Ling Yang

**Affiliations:** 1Department of Biology, Shenzhen MSU-BIT University, Shenzhen 518172, China; kitashov@smbu.edu.cn; 2College of Forestry, Beijing Forestry University, Beijing 100083, China; 3State Key Laboratory of Tree Genetics and Breeding, Northeast Forestry University, Harbin 150040, China; zhangyue19970309@163.com (Y.Z.); wangmengfan61@163.com (M.W.); 4Biological Faculty, Lomonosov Moscow State University, Moscow 119991, Russia

**Keywords:** *ASR* genes, development history, gene structure, fruit ripening, abiotic stress, biotic stress

## Abstract

Abiotic and biotic stress factors seriously affect plant growth and development. The process of plant response to abiotic stress involves the synergistic action of multiple resistance genes. The *ASR* (*Abscisic acid stress-ripening*) gene is a plant-specific transcription factor that plays a central role in regulating plant senescence, fruit ripening, and response to abiotic stress. *ASR* family members are highly conserved in plant evolution and contain ABA/WBS domains. *ASR* was first identified and characterized in tomatoes (*Solanum lycopersicum* L.). Subsequently, the *ASR* gene has been reported in many plant species, extending from gymnosperms to monocots and dicots, but lacks orthologues in *Arabidopsis* (*Arabidopsis thaliana*). The promoter regions of *ASR* genes in most species contain light-responsive elements, phytohormone-responsive elements, and abiotic stress-responsive elements. In addition, *ASR* genes can respond to biotic stresses via regulating the expression of defense genes in various plants. This review comprehensively summarizes the evolutionary history, gene and protein structures, and functions of the *ASR* gene family members in plant responses to salt stress, low temperature stress, pathogen stress, drought stress, and metal ions, which will provide valuable references for breeding high-yielding and stress-resistant plant varieties.

## 1. Introduction

The natural environment in which plants exist is dynamic, with stress conditions such as extreme temperatures, drought, salinization, fungi, and other biotic stress factors significantly impacting plant growth and development [1]. Plant hormones play a crucial role in regulating plant growth, development, and stress responses, enhancing plant tolerance to various stresses [2]. This regulatory effect is not the result of a single action, but is mostly linked through some common elements, commonly referred to as hubs or links of crosstalk [3]. ABA (abscisic acid) is a significant plant hormone that regulates plant growth, development, and stress responses when interacting with other plant hormones [4]. The crosstalk between ABA and GA (gibberellin) can modulate seed dormancy, germination, maturation, root growth, and flowering time [5]. ABA can also influence the expression of IAA (auxin) response factors *ARF5*, *ARF6*, and *ARF10* through ubiquitination responses, thereby controlling the IAA signaling pathway and enhancing plant tolerance to salt stress [6]. Additionally, ABA is involved in promoting fruit ripening [7] and increasing fruit yield [8]. The interaction between ABA and CKs (cytokinin) is particularly evident in breaking seed dormancy [9], inhibiting bud dormancy and leaf senescence [10], as well as enhancing drought tolerance in plants [11,12]. The relationship between ABA and ETH (ethylene) can be both positive and negative, depending on the specific developmental process. ABA can regulate ETH biosynthesis and signaling through controlling the expression of *ethylene response factor 11*, *NCED*, and *acyl-CoA*. For example, ABA can induce ETH biosynthesis in the regulation of photoperiod, fruit ripening, and leaf senescence [13]. Conversely, in the case of long-term darkness, ABA inhibits flowering via suppressing ETH biosynthesis [14]. Additionally, the interaction between ABA and SA (salicylic acid) is particularly significant in responses to drought stress, heavy metal stress, and biotic stress [15,16,17]. The role of JA (jasmonic acid) in the stress response is similar to that of ABA, with their interaction mainly reflected in processes such as triggering stomatal closure and leaf senescence to regulate plant tolerance to drought and low temperatures [18,19]. Furthermore, the stomatal closure promoted by ABA is antagonistically regulated by brassinosteroids and SA, while promoting leaf senescence via counteracting the effects of melatonin and CKs [20].

*ASR* genes, plant-specific TFs (transcription factors), are crucial in regulating plant senescence, fruit ripening, and response to abiotic stress [21]. These genes are expressed in plant organs and growth stages, playing a significant role in plant response to developmental and environmental conditions, particularly involving ABA signaling [22,23,24,25,26,27]. ASR proteins exhibit strong hydrophilicity, indicating a potential role in drought stress response. Since the first *ASR* gene was identified in tomatoes, *ASR* genes have been identified in various monocotyledonous and dicotyledonous plants. However, they have not been found in the model plant *Arabidopsis* [28]. Studies have shown that ASR proteins typically function as molecular chaperones, osmoregulatory proteins, metal-binding proteins, and antioxidant or detoxification proteins [29,30,31]. Some ASRs also act as unique transcription factors [32]. Therefore, exploring the functions of ASRs in plant growth, development, fruit ripening, and stress responses is crucial for enhancing economic production.

## 2. The Role of ABA in Plant Growth and Stress Response

Plant growth and development can be influenced by various stress factors in the natural environment. These factors include abiotic stress such as temperature fluctuations, lack of water and nutrients, and soil salinization, as well as biotic stress factors such as infestation via fungi, bacteria, and herbivores. Investigating the mechanism of stress tolerance in plants is crucial. Studies have shown that biotic and abiotic stresses usually occur simultaneously in nature and that there is crosstalk in the plant responses to these stresses, with ABA being the central component of this crosstalk [3,33,34].

ABA was first identified and defined as a plant growth inhibitor in the 1860s. ABA plays an important role in the regulation of plant growth, stress response, and various physiological processes, including seed dormancy and germination [35,36,37,38,39], stomatal movement [40,41,42], fruit development [43,44,45,46], and the response to biotic and abiotic stress factors [47,48,49,50]. Seed dormancy is the main mechanism for plants to overcome stress [51], and ABA plays a significant role in seed dormancy and development. ABA induces seed dormancy and maturation and inhibits seed germination via regulating the activity of transcription factors such as *LEC1* (*Leafy Cotyledon1*) and *LEC2* (*Leafy Cotyledon2*). Excessive accumulation of *FUSCA3* and *ABI3* (*Abscisic Acid Insensitive3*) can also inhibit seed germination [52,53]. Additionally, it has been shown that ABA synthesis in the endosperm is the main reason for seed dormancy [37,54]. It was found that the interaction between *ODR1* and *bHLH57* prevented binding to the *NCED6* and *NCED9* promoters, which in turn affected ABA biosynthesis and downregulated the transcription of *ABI4*. During seed maturation, binding of *ABI3* to the *ODR1* promoter represses its expression and releases the repressive effects of *bHLH57*, thereby promoting ABA biosynthesis and seed dormancy [54]. ABA inhibits seed germination through preventing seed cracking and endosperm rupture [35]. ABA also influences the growth of plant roots. It was found that reduced endodermal ABA signaling decreased miR165 levels and led to a significant increase in ATHB8 and REV transcript levels, which in turn affected xylem development [55]. Besides forming xylem through microRNA and its target transcription factors, ABA is a key signaling molecule for plants to establish a hydrophobic suberin protein barrier to prevent water and nutrient outflow [56], and it supports the transport of nutrients and water from the xylem to the buds. Furthermore, ABA can promote the growth of primary roots via reducing the biosynthesis of ETH, playing a role in maintaining the root meristem during primary root growth [57]. Exogenous ABA has been shown to promote the ripening of plants such as tomatoes, bananas, peaches, mangoes, and melons through regulating several physiological mechanisms associated with ripening [58,59,60,61]. As an upstream regulator of ETH biosynthesis and signal transduction, ABA positively regulates ETH synthesis during fruit ripening [62,63]. The endogenous ethylene induces the expression of *VvNCED1*, encoding 9-cis-epoxycarotenoid dioxygenase (NCED) and *VvGT*, thereby encoding an ABA glucosyltransferase; both increased rapidly during grape ripening. When the level of ABA reaches the peak value, part of it will be stored in the form of ABA-GE. Their interaction thus initiates the berry ripening process [62]. The exogenous addition of ABA also promotes ETH synthesis and accelerates the plant respiration rate to promote fruit ripening. However, another study showed that the ETH production of the ABA-deficient mutant flc was significantly higher than that of the wild type. This indicates that ABA can inhibit the production of excess ETH, suggesting that ABA’s regulation of ETH can be positive or negative depending on the tissue type or developmental stage [64]. In addition, ABA levels were negatively correlated with stomatal density and index. ABA function deficiency mutants (*aba1*, *aba2*, *aba3*, *nced3*, *nced5*, *atbg1*, and *atbg2*) of *Arabidopsis* showed an increase in stomatal density and stomatal pressure. In contrast, *cyp707a1* and *cyp707a3* mutants, which lack the catabolic function of ABA, showed a decrease in stomatal density and index [65,66,67,68]. ABA can also reduce the size of guard cells, thus affecting the size of plant epidermal cells [66,69]. This suggests that ABA is involved in the process of plant stress response through influencing the development of stomata and guard cells.

The most studied function of ABA is its role in abiotic stress responses such as salt, drought, and low temperature. Especially under drought stress, plant cells rapidly accumulate a large amount of ABA. Under drought stress, the ABA content in the leaves increases tenfold or more [70]. Even under long-term drought stress, the synthesis and catabolism rates of ABA in *Arabidopsis* and soybean (*Glycine max*) are still high. Moreover, ABA accumulated in guard cells under drought stress promotes stomatal closure and inhibits stomatal opening to reduce transpiration-induced water loss [71,72,73]. ABA can also cause a transient increase in intracellular calcium levels, thereby activating slow-activating (S-type) and fast-activating (R-type) anion channels at the plasma membrane [74]. This suggests that the rapid renewal and accumulation of ABA are the most important mechanisms for plants to cope with drought stress. Moreover, ABA content in plants is also significantly increased under a low temperature and salt stress [75,76,77,78,79,80], and the exogenous application of ABA can improve plant resistance to abiotic stresses such as drought [81,82,83], high salinity [84,85,86], and low temperatures [87,88]. ABA regulates the expression of many genes in plants in response to abiotic stresses such as drought, high salinity, and low temperatures, with several types of transcription factors involved in the ABA-mediated regulation of gene expression. For example, CaSnRK2.4 interacted with and phosphorylated CaNAC035 in vitro and in vivo. In addition, the expression of two ABA biosynthesis-related genes (*CaAAO3* and *CaNCED3*) was significantly upregulated in transgenic pepper (*Capsicum annuum*) lines with high expression of *CaNAC035*, which enhanced tolerance to cold stress [87]. These transcription factors interact with cis-acting elements in the promoter region of ABA-induced genes to activate their expression. The cis-acting element *ABRE* and the dehydration-responsive element DRE are the major stress response elements that play a role in ABA-dependent and ABA-independent gene expression, respectively. In *Arabidopsis*, *NCED3* is considered the major determinant of ABA accumulation under drought stress [89], promoting ABA accumulation and stomatal closure [90]. Additionally, the ABA-responsive genes *RD29A*, *RD29B*, *AREB1/ABF2*, *ABF3*, and *AREB/ABF4* also play an important role in plant drought tolerance [91]. *LT178, COR78, RD29A,* and NACs are significantly induced under high salinity, drought, and low-temperature stress, which are essential for the abiotic stress responses in the plants [92]. Heat shock factor *HSFA6b* is activated by ABA-mediated *AREB1* and positively regulates drought and high-temperature stress [93]. Furthermore, *RD26* is the main signal molecule for plant response to low-temperature stress [92].

## 3. The Development History of *ASR* Gene Family Members

*ASR* gene members contain a conserved ABA/WDS structural domain. *ASR* genes encode small, plant-specific hydrophilic proteins that are not only involved in plant responses to drought, high salinity, low temperature, and ABA stresses and are closely related to the ABA signaling pathway, but also play a role in many plants’ metabolic processes, such as fruit ripening and sugar metabolism. The first *ASR* gene, *ASR1*, was first identified in tomatoes (*Solanum lycopersicum* L.) in 1993 via screening differential genes in tomato leaves and ripe fruits under drought stress [94]. In 1994, another *ASR* gene with high sequence similarity to *ASR1* was cloned and named *ASR2* [95]. Further studies revealed that *ASR1* and *ASR2* are part of a gene family with at least three members, namely *ASR1*, *ASR2*, and *ASR3*, which are clustered on chromosome 4 of tomatoes [96]. Subsequently, *ASR* genes have been found in papayas (*Pseudocydonia sinensis*), tobacco (*Nicotiana tabacum* L.), and potatoes (*Solanum tuberosum* L.) [97]. In 2006, the fourth *ASR* gene was identified and named *ASR4*. This gene is highly similar in structure to the previously identified *ASR* gene but contains a specialized structural domain and positively responds to drought stress [98]. The fifth *ASR* gene, *ASR5,* was identified in 2011. *ASR5* and *ASR3* have highly similar coding regions, but their intron sequences are very different. This study also showed that individual members of the *ASR* gene family exhibited diverse evolutionary histories [99]. Meanwhile, *ASR* genes have also been identified and cloned in various species, including the potato [100], maize (*Zea mays* L.) [101], pummelo (*Citrus maxima*) [102], loblolly pine (*Pinus taeda*) [103], lily (*Lilium longiflorum Thunb*. cv.) [104], rice (*Oryza sativa* L.) [105], grape (*Vitis vinifera* L.) [106], apple (*Malus* × *domestica*) [107], chickpea (*Cicer arietinum* L.) [108], cucumber (*Cucumis sativus*) [109], and peach (*Prunus persica* f. atropurpurea) [110]. Functional analysis of the *ASR* gene has demonstrated their association with biological processes such as seed dormancy and germination [111], fruit development and ripening [112], and stress response [107,108]. Currently, *ASR* genes have been identified and characterized in many species through whole genomes, including 6 members in rice [113], 5 members in tomato [98], 4 members in loblolly pine [103], 9 members in maize [101], 33 members in wheat (*Triticum aestivum* L.) [114], 5 members in *Brachypodium* (*Brachypodium distachyon*) [115], 10 members in bay bean [116], 5 members in apple [107], and 4 members in banana (*Musa nana* Lour.) [117]. There is 1 member in grape [106] and 27 members in 8 Rosaceae (Table 1) [118]. However, no homology to the *ASR* gene was found in the model plant *Arabidopsis*, making it difficult to study the function of the *ASR* gene using *Arabidopsis* mutants [119].

## 4. Structure, Physicochemical Properties, and Expression Patterns of *ASR*

The *ASR* genes of most species, including wheat [114], tomato [123], banana [117], maize [121], bay bean [116], and apple [107], contain two exons (Table 1), while the exons of *ASR* genes in Rosaceae plants’ exons usually number between one and four [118]. The promoter elements of *ASR* genes in most species contain light-response elements, ABA-response elements, GA-response elements, MeJA (methyl jasmonate)-response elements, IAA-response elements, SA-response elements, drought-response elements, and low-temperature-response elements [114,116,118,121]. This suggests that *ASR* may interact with hormone signaling networks to regulate plant growth and stress response. In addition, the results of subcellular localization showed that all *ASR* genes are localized in the nucleus [114,115,116,121,124]. Duplication is the main driver of gene expansion during the evolution of species. Five types of gene duplication can occur during evolution, including singleton, dispersed, tandem, proximal, and segmental duplication. The results showed that among the 29 *ASR* genes in wheat, 29 pairs of tandem duplication and 12 pairs of segmental duplication had emerged, indicating that tandem duplication and segmental duplication were the main driving forces of *ASR* evolution. In addition, Ka/Ks can be used to determine whether selective pressure is acting on protein-coding genes (Ka/Ks < 1 for purifying selection, Ka/Ks = 1 for neutral selection, and Ka/Ks > 1 for positive selection). The *ASR* genes of different species were subject to different environmental selections during the evolutionary process. For example, *ASR1* of the tomato was subject to environmental purifying selection while *ASR2* was subject to positive selection, and *ASR1*, *ASR2*, and *ASR3* of the apple were subject to environmental purifying selection while *ASR4* and *ASR5* were subject to positive selection. These results indicate that the *ASR* genes in tomatoes and apples have evolved mainly to adapt to the environment. In contrast, all *ASR* genes in Rosaceae were subject to purifying selection by the environment, suggesting that the *ASR* genes in Rosaceae are highly conserved during evolution [107,118,125]. Furthermore, the ASRs of all species contain conserved ABA/WDS protein domains and are hydrophilic proteins. The stability of ASR proteins differs between species. For example, the ASR proteins of beans and Rosaceae were thermally stable [116,118], while the ASR proteins of apples were unstable [107].

*ASR* has been shown to be essential for plant growth and stress response, and the roles of different *ASR* members are quite diverse. *SlASR1* was highly expressed in tomato roots and stems and promoted the development of nutrient organs [123]. The expression levels of *ASR3* and *ASR5* were highest in tomato cotyledons, while *ASR4* was highest in mature leaves [123]. Similar to the tomato, most *ASR* genes in Rosaceae have the highest expression levels in leaves and fruits [118]. Of the 33 *ASR* genes in wheat, 24 were expressed in roots, stems, leaves, flowers, and fruits [114]. Five *ASR* genes of bay bean were expressed in all plant tissues, indicating that *ASR* genes are essential for plant growth and development [115]. In addition, the *ASR* genes responded to salt stress, drought stress, low-temperature stress, high-temperature stress, and pathogen infection [107,114,115,116,121,123], and were significantly upregulated after treatment with the phytohormone ABA [123]. Similar to the results of the promoter element analysis, this suggests that *ASR* is linked to the ABA signaling network to regulate plant growth and development as well as the stress response process.

## 5. *ASR* Regulates Fruit Ripening

*ASR* plays a crucial role in fruit ripening through modulating the synthesis and metabolism of glucose, cell wall components, amino acids, ABA, and carotenoids. Studies have shown that *ASR* is situated within the signaling cascade involving glucose, ABA, and GA. Reduction in ASR protein levels results in the diminished activity of tobacco HK1 (Hexokinase1), impacting glucose metabolism, photosynthesis, and respiration. This reduction also leads to decreased levels of ABA and GA, contributing to plant dwarfism and hastening leaf senescence (Table 2) [125].

Overexpression of plum *PpASR1* in tomatoes has been shown to regulate the metabolic processes of the tomato cell wall, leading to a significant increase in the levels of anthocyanin and lycopene [110]. Similarly, overexpression of *ASR1* in strawberries significantly increased the expression levels of anthocyanin biosynthesis-related genes *CHS* (chalcone synthase), *CHI* (chalcone isomerase), *ANS* (anthocyanin synthase), and the ABA biosynthesis gene *NCED* in strawberries and tomatoes. This promotion of anthocyanin and ABA synthesis further enhances the ripening and softening processes in both strawberry and tomato fruits [126]. *ASR* can also regulate fruit ripening through regulating amino acid metabolism. *ZmASR1* has been shown to impact the biosynthesis of BCAA (branched-chain amino acid) and regulate maize grain yield [101]. Tomato *ASR1* has been found to enhance the accumulation of proline, methionine, valine, and isoleucine in fruits, thereby promoting fruit development [21,124]. Furthermore, acting as a downstream component of the ABA signaling pathway, *ASR* has been implicated in the regulation of strawberry fruit ripening and firmness by modulating ABA levels. It has also been linked to an increase in the number of tillers and grain yield in rice [21]. Recent transcriptome and metabolomics analyses have revealed that *ASR1* can affect plant photosynthesis, tricarboxylic acid cycle (citrate and succinate), lipid metabolism (phenylalanine and glycerol), and isoprene synthesis (β-carotene and lycopene) pathways, which are also closely related to plant growth and fruit ripening [21].

## 6. Response of *ASR* to Low-Temperature Stress

Currently, a large number of studies focus on the mechanism of *ASR* response to low-temperature stress, including the functions of *ASR* genes in rice, maize, and lily. The results showed that low-temperature stress significantly induced the expression of *OsASR1* in the vegetative and reproductive organs of rice, and overexpression of *OsASR1* significantly increased the photosynthetic efficiency (Fv/Fm) of rice leaves under low-temperature stress (Figure 1) [127].

## 7. Response of *ASR* to Metal Ions

Overexpression of *OsASR3* can also enhance the tolerance of rice to low-temperature stress [128]. Similarly, overexpression of *ZmASR1* in maize has been shown to improve photosynthesis efficiency, reduce lipid peroxidation levels, and increase the activities of antioxidant enzymes under low-temperature stress (Figure 1) [129]. Heterologous expression of the *ASR* family member *LLA23* in *Arabidopsis* promoted the growth of stems and roots via preventing electrolyte leakage, inducing the expression of genes related to low-temperature stress and enhancing the activity of cryoprotective enzymes MDH and LDH (Figure 1) [130]. Furthermore, *TtASR1* in *E. coli* also improved the tolerance of *E. coli* to low-temperature stress [112]. The expression profile of *ASR* genes under low-temperature stress has also been characterized in many species. In rice, for instance, the expression of all *ASR* family members *OsASR1*–6 was found to be significantly upregulated in response to low-temperature stress [131]. *BdASR4*, *BdASR1*, *BdASR2*, *BdASR3*, and *BdASR5* exhibited positive responses to low-temperature stress [115]. Among the 29 *ASR* genes in wheat, 11 *ASR* genes showed a positive response to low-temperature stress [122]. Conversely, the expression level of *DiASR1* in the dove tree was significantly downregulated under low-temperature stress, indicating that the function of *ASR* genes in response to low temperatures varies among different species (Figure 1) [132]. Furthermore, the ABA/WDS domain may play a crucial role in the response of *ASR* genes to low-temperature stress and other abiotic stresses [133].

Metal ions have a negative impact on plant growth and development (Figure 1) [134,135]. The structure and function of ASR proteins can be regulated through binding to metal ions. Most ASR proteins contain a Zn^2+^ binding domain. As a chaperone-like protein in the cytoplasm, tomato SlASR1 has a Zn^2+^-dependent binding activity. Binding to Zn^2+^ led to the transformation of the protein structure of SlASR1 from an unfolded, disordered state to a folded, ordered state and induced the DNA-binding activity of SlASR1 [136,137]. Under abiotic stress, SlASR1 binds to Zn^2+^ in the cytoplasm and excretes Zn^2+^ out of the cell (Figure 1) [138]. Similar to tomato SlASR1, the protein structure of TtASR1 and HvASR1 also changed from an unfolded, disordered state to a folded, ordered state after Zn^2+^ treatment (Figure 1) [138]. Rice exhibits the highest tolerance to aluminum toxicity among all cereal plants. The expression level of the *OsASR5* gene is notably higher when compared with aluminum-sensitive rice genotypes. After 4 to 8 h of aluminum stress, the expression levels of *OsASR1*–*6* varied in their degrees of increase. *OsASR5* RNAi rice demonstrated significantly higher aluminum sensitivity than wild-type rice, resulting in delayed flowering, abnormal panicle shape development, and grain loss (Figure 1) [139]. *OsASR5* and *OsASR1* have complementary functions in response to aluminum stress. *OsASR5* regulates the expression of 36 aluminum response genes, including *STAR1* and *STAR2*. This suggests that under aluminum stress, the transcript levels of *OsASR5* are significantly elevated, leading to a positive regulation of *OsASR1* expression. Additionally, the *STAR1/STAR2* complex, along with *OsASR1*, works together to cover the aluminum binding sites in the cell wall [140,141]. The expression level of *ZmASR1* in the roots, stems, and leaves of maize was significantly increased under cadmium stress, and tobacco and yeast overexpressing *ZmASR1* were found to be tolerant to cadmium stress (Figure 1) [140]. It has been shown that GmASR proteins in soybean with Fe^3+^, Ni^2+^, Cu^2+^, and Zn^2+^ binding sites could prevent oxidative damage through buffering metal ions, thus alleviating metal toxicity in plant cells under stress conditions (Figure 1) [142].

## 8. Response of *ASR* to Salt Stress

Under salt stress, the excessive accumulation of Na^+^ can lead to ion imbalance, oxidative stress, nutrient deficiency, growth retardation, and cell death [143]. Studies have demonstrated that *ASR* responds to salt stress via eliminating excess ROS, activating stress response genes, and regulating intracellular Na^+^/K^+^. The expression level of the *ThASR3* gene in *Tamarix hispida* was significantly increased, which was similar to the expression patterns of *OsASR6* in rice, *CrASR* in bay bean, *HvASR5* in barley, and *TaASR1-D* in wheat, indicating that a conserved role in *ASR* family members across different species is in their salt response (Figure 1) [116,144,145,146]. Salt stress triggered a surge in ROS levels within cells, disrupting the redox balance. Overexpression of *ASR* genes can boost antioxidant enzyme activity, enhance the accumulation of osmotic regulators like proline and glycine betaine, reduce intracellular MDA content and electrolyte permeability, improve ROS scavenging capacity, stimulate root, stem, leaf, and fruit development, and ultimately increase the yield of plants [144,146,147,148]. *ASR* can also enhance salt stress resistance via upregulating the expression of salt-responsive genes. Overexpression of *BdASR2* in *Brachypodium* resulted in increased expression levels of *BdWRKY36*, *BdSOS2*, and *BdHKT7*, as well as antioxidant enzyme genes *BdCAT*, *BdAPX2* and *BdMn-SOD* [148]. Similarly, the expression of antioxidant enzyme genes *APX2*, *FSD1*, *CSD1*, and *CAT1* was also significantly induced in *Arabidopsis* heterologously expressing *IpASR* (Figure 1) [147], suggesting that *ASR* responded to salt stress via controlling ROS homeostasis and regulating the expression of stress-related genes. Banana *MaASR1* was found to improve salt tolerance in *Arabidopsis* through downregulating the ABA-dependent stress response genes, while leaving the ABA-independent genes and ABA biosynthesis pathway genes unaffected (Figure 1) [149]. Moreover, *ASR* can reduce the Na^+^/K^+^ ratio via reducing Na^+^ uptake and eliminating excess Na^+^ from cells [150]. For instance, overexpression of *OsASR6* in rice and *TaASR1-D* in wheat led to a significant reduction in Na^+^/K^+^ levels in leaf cells and seedlings, respectively, indicating that *ASR* plays a crucial role in maintaining ion homeostasis both internally and externally (Figure 1) [26,144]. In addition, studies have demonstrated that heterologous expression of the *ASR* genes enhanced salt stress tolerance in yeast and *E. coli* [109,112,116,151]. A recent study has demonstrated that *ASR* was mainly involved in chitin catabolism, redox balance, cell wall modification, defense response to fungi, defense response to low-temperature stress, and transcriptional regulation [145].

## 9. Response of *ASR* to Drought Stress

Drought stress significantly affects plant growth and reduces crop yield. Plants have evolved complex mechanisms to cope with drought stress. Studies have shown that *ASR* responds to drought stress via scavenging excessive ROS in cells, inducing stress-responsive gene expression, promoting compatible solute accumulation, and regulating stomatal morphology. Similarly, drought stress also leads to a high accumulation of intracellular ROS and disrupts the redox balance. Overexpression of *ASR* genes can increase the activities of antioxidant enzymes, such as SOD, POD, CAT, and GSH, and scavenge excessive H_2_O_2_ and MDA in cells to maintain the redox balance [111,113,115,151,152,153]. *ASR* can also promote the accumulation of free amino acids such as proline in tissues to resist drought stress [110,151]. Drought stress can result in damage to cell membrane and cell wall structures, leading to the outflow of cell solutes. Heterologous expression of *PpASR* gene in tobacco significantly reduced the ion outflow of tobacco cells, which was essential for the stability of cell structure under drought stress [110]. *SiASR1* from foxtail millet (*Setaria italica*), *TaASR1* from wheat, and *OsASR6* from rice were also shown to reduce cell outflow (Figure 1) [111,113,151]. Drought stress is recognized for triggering the activation of stress response genes. Heterologous expression of *Brachypodium BdASR1* in tobacco led to a significant increase in the expression of stress-related genes such as *NtEDR10C*, *NtEDR10D*, *NtLTP1*, and *NtDREB3*, as well as genes related to the synthesis of antioxidant enzymes such as *NtSOD*, *NtPOX*, and *NtCAT* (Figure 1) [115]. Similarly, heterologous expression of *SiASR1* from *Setaria italica* in tobacco also resulted in significantly increased expression levels of the antioxidant enzyme synthesis genes *SOD*, *POD*, and *CAT* (Figure 1) [113]. The heterologous expression of *ZmASR3* from maize in *Arabidopsis* led to a significant induction of the stress-resistant genes *AtCOR2* and *AtDREB15A* (Figure 1) [153]. The expression of stress-responsive genes such as *CBF1*, *CBF2*, *CBF3*, *DREB2A*, *RAB16*, *WRKY36*, *AREB*, *LEA*, *NCED2*, and *P5CS1* was also shown to be positively regulated in wheat *TaASR2D* and moso bamboo *PheASR2* (Figure 1) [152,154]. This suggests that *ASR* responds to drought stress via controlling ROS homeostasis and regulating the expression of stress-related genes. Furthermore, it was found that the positive regulation of *ASR* during drought stress can be counterbalanced by exogenous ABA. Overexpression of *ZmASR3* in plants led to an increased accumulation of ABA in leaves, resulting in a significant increase in the expression levels of ABA-dependent genes including *NCED3*, *AAO3*, *EDR1*, *RAB18*, and *SnRK2.6* [152,153,154]. Drought stress typically causes changes in stomatal structure and density. *ASR* has been found to decrease stomatal opening, stomatal density, and stomatal conductance in leaves under drought conditions, ultimately reducing leaf water loss. This process is closely related to the ABA signaling pathway and is finely regulated by ABA. These findings suggest that *ASR* modulates stomatal structure and density through an ABA-dependent signaling pathway, playing a crucial role in plant responses to drought stress [30,146,153].

## 10. Response of *ASR* to Pathogen Infection

The role of *ASR* in abiotic stress has been extensively studied, but there are few reports on its role in biotic stress. Studies have shown that the expression of *OsASR6* in rice was significantly induced after *Xoo* (*Xanthomonas oryzae* pv. *oryzae*) and *Xoc* (*Xanthomonas oryzae* pv. *oryzicola*) treatments. The expression levels of *MpASR* in plantain (*Musa paradisiaca*) increased significantly after treatment with *Fusarium oxysporum* f. sp. *Cubense*, suggesting that they may act as positive regulators in response to pathogens (Figure 1) [155,156]. The expression levels of *MdASR2*, *MdASR3*, and *MdASR6* were consistently downregulated after inoculation. This indicates that they respond to the negative regulator of *Alternaria alternata* f. sp. *Mali* (Figure 1) [107]. The *ASR* gene has been shown to respond to biotic stress via regulating the expression of defense genes. Specifically, *OsASR2* was found to regulate the activity of the GT-1 element in the promoter of *Os2H16* and enhance the resistance to *Xoo* and *Rhizoctonia solani* (Figure 1) [157]. Interestingly, *OsASR6*, another member of the rice *ASR* family, regulates the expression of target genes *PibHs*, *ASN1*, *WRKYs*, and *CIPKs*, thereby affecting rice resistance to *Xoo* and *Xoc* [156], suggesting that different members of the *ASR* family within the same species may have varying functions in response to stress. Furthermore, tomato *ASR1* positively influences the infection of the necrotrophic fungus *Botrytis cinerea* [21]. In addition to regulating the expression of defense genes, *ASR* enhances resistance to pathogens through promoting the activity of antioxidant enzymes and defense-related enzymes PPO and PAL, as well as through reducing ROS accumulation [107]. A recent study has demonstrated that pepper *CaASR1* can enhance pepper resistance to *Capsicum annuum* through promoting SA-dependent signal transduction and inhibiting JA-dependent signal transduction. These findings suggest that the SA pathway may serve as the central pathway for ASR-mediated pathogen response (Figure 1) [158].

## 11. Conclusions

*ASR* plays a crucial role in promoting plant growth and fruit ripening through regulating the biosynthesis and metabolism of various compounds such as glucose, cell wall components, amino acids, ABA, and carotenoids, ultimately leading to increased fruit and grain yield. Currently, research on *ASR* promoting fruit ripening mainly focuses on Rosaceae and Solanaceae, as well as cereal crops.

Plant responses to abiotic stresses can be attributed to reducing the damage caused by osmotic stress, such as through promoting the activity of antioxidant enzymes, accelerating intracellular Na^+^ efflux, scavenging excessive ROS, inducing the expression of antioxidant enzyme genes, and promoting the accumulation of osmotic protectants and free amino acids. Additionally, this response is closely related to the ABA signal transduction pathway. *ASR* responds to biotic stress through controlling ROS homeostasis and inducing the expression of downstream stress-related genes. Among these, the SA pathway may be the core pathway of the *ASR*-mediated biotic stress response.

Although the role of *ASR* in fruit ripening and abiotic stress has been extensively studied, its role in biotic stress has rarely been reported. Considering the importance of cereal crops in the development of national economy and the susceptibility of cereal pathogens under natural conditions, exploring the mechanism of the *ASR* gene response to biotic stress should be the main direction of *ASR* functional analyses.

## Figures and Tables

**Figure 1 ijms-25-10283-f001:**
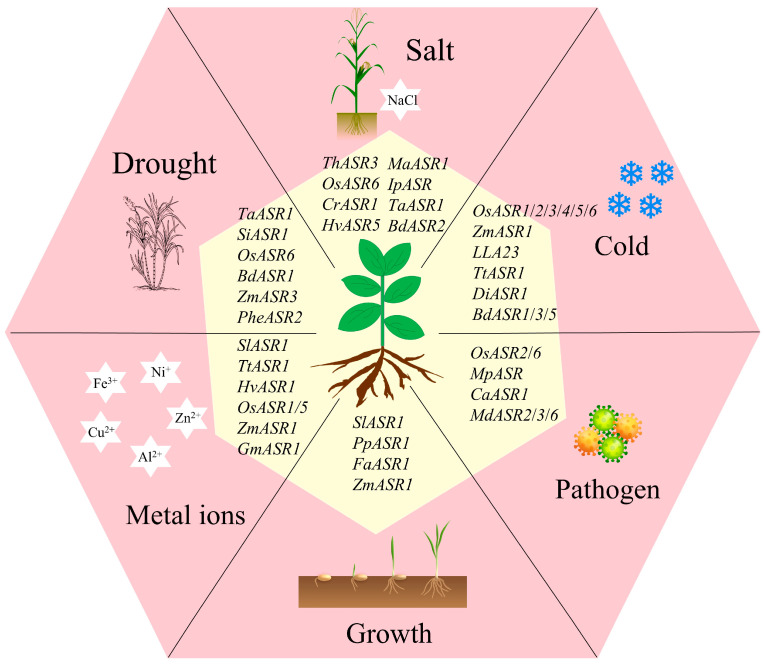
*ASR* genes responding to low-temperature stress, drought stress, pathogen stress and metal ions, and as well as being involved in plant growth. The external pink areas represents plant growth and response to different stress conditions. The internal yellow areas represent *ASR* genes in different species.

**Table 1 ijms-25-10283-t001:** Gene structure, subcellular localization, and protein physicochemical properties of ASRs from different species. References: [107,113,115,116,118,120,121,122].

Species	Gene ID	Exons	Introns	Length (aa)	Mw (kD)	PI	Gravy	Subcellular Localization
*Canavalia rosea*	*CrASR1*	2	1	235	25.92	5.79	−1.449	Nucleus, Cytoplasm
	*CrASR2*	2	1	130	14.98	6.34	−1.321	Nucleus, Cytoplasm, Mitochondria
	*CrASR3*	2	1	114	13.09	6.41	−1.341	Nucleus, Cytoplasm, Mitochondria
*Triticum* *aestivum*	*TaASR1*	2	1	137	15.3	6.06	−1.2	Nucleus
	*TaASR2*	2	1	264	28.83	5.19	−1.76	Nucleus
	*TaASR3*	2	1	110	12.34	9.89	−1.29	Nucleus, Cytoplasm
	*TaASR4*	2	1	97	10.76	9.99	−1.33	Nucleus
	*TaASR5*	2	1	110	12.34	9.89	−1.29	Nucleus
	*TaASR6*	2	1	94	10.39	9.82	−1.23	Nucleus
	*TaASR7*	2	1	219	23.18	6.25	−0.98	Nucleus
	*TaASR8*	2	1	218	23.21	6.24	−0.93	Nucleus
	*TaASR9*	2	1	219	23.68	6.16	−1.07	Nucleus
	*TaASR10*	2	1	230	24.91	7.79	−0.99	Nucleus
	*TaASR11*	2	1	110	12.34	9.89	−1.29	Nucleus
	*TaASR12*	3	2	134	15	6.11	−1.04	Nucleus, Cytoplasm
	*TaASR13*	2	1	279	30.34	4.97	−1.76	Nucleus
	*TaASR14*	2	1	100	11.04	9.99	−1.35	Nucleus, Cytoplasm
	*TaASR15*	2	1	227	24.39	6.27	−1.09	Nucleus, Cytoplasm
	*TaASR16*	2	1	221	23.45	6.1	−0.95	Nucleus, Cytoplasm
	*TaASR17*	2	1	219	23.14	6.03	−0.96	Nucleus
	*TaASR18*	2	1	97	10.84	9.99	−1.33	Nucleus
	*TaASR19*	2	1	138	15.46	6.14	−1.2	Nucleus
	*TaASR20*	2	1	262	28.65	5.2	−1.74	Nucleus, Cytoplasm
	*TaASR21*	3	2	175	18.86	6.51	−1.11	Nucleus
	*TaASR22*	2	1	218	23.2	6.24	−0.92	Nucleus, Cytoplasm
	*TaASR23*	2	1	220	23.25	6.19	−0.99	Nucleus, Cytoplasm
	*TaASR24*	2	1	94	10.44	9.7	−1.22	Nucleus, Cytoplasm
	*TaASR25*	2	1	91	10.1	9.87	−1.21	Nucleus
	*TaASR26*	2	1	94	10.37	9.74	−1.23	Nucleus, Cytoplasm
	*TaASR27*	2	1	97	10.81	10.04	−1.37	Nucleus, Cytoplasm
	*TaASR28*	2	1	100	11.16	10.14	−1.33	Nucleus, Cytoplasm
	*TaASR29*	2	1	110	12.34	9.89	−1.29	Nucleus
*Setaria italica*	*SiASR1*	1	2	200	22.5	6.3	−1.190	Nucleus
	*SiASR2*	1	1	137	15.4	6.2	−1.277	Nucleus
	*SiASR3*	2	1	105	11.7	9.7	−1.147	Nucleus
	*SiASR4*	3	2	102	11.5	9.3	−1.417	Nucleus
	*SiASR5*	1	1	173	19.4	6.3	−1.366	Nucleus
	*SiASR6*	2	1	101	11.5	6.8	−1.636	Nucleus
*Brachypodium distachyon*	*BdASR1*	2	1	201	21.82	6.18	−1.061	Nucleus
	*BdASR2*	2	1	111	12.27	9.84	−1.155	Whole cells
	*BdASR3*	2	1	102	11.05	9.82	−1.208	Whole cells
	*BdASR4*	2	1	139	15.37	6.17	−1.025	Whole cells
	*BdASR5*	2	1	240	25.97	5.11	−1.641	Whole cells
*Pyrus bretschneideri*	*PbrASR1*	2	1	203	22.35	5.63	−1.426	Nucleus
	*PbrASR2*	2	1	182	19.81	5.96	−1.387	Nucleus
	*PbrASR3*	2	1	134	15.29	6.1	−1.439	Nucleus, Cytoplasm
*Fragaria vesca*	*FvASR1*	2	1	192	21.02	6.03	−1.278	Nucleus
*Malus × domestica*	*MdASR1*	2	1	202	22.12	5.74	−1.421	Nucleus
	*MdASR2*	2	1	133	12.05	6.17	−1.276	Nucleus
	*MdASR3*	2	1	119	13.51	9.69	−0.639	Nucleus
*Prunus avium*	*PavvASR1*	2	1	135	16.41	5.36	−1.082	Nucleus
	*PavvASR2*	2	1	283	29.99	5.39	−1.54	Nucleus
	*PavvASR3*	2	1	277	29.35	5.53	−1.537	Nucleus
	*PavvASR4*	2	1	269	28.06	5.53	−1.271	Nucleus
	*PavvASR5*	2	1	276	29.15	5.13	−1.523	Nucleus
	*PavvASR6*	2	1	66	7.7	6.18	−1.432	Nucleus
	*PavvASR7*	2	1	66	7.73	6.24	−1.538	Nucleus
	*PavvASR8*	2	1	110	12.57	6.26	−1.328	Nucleus
*Pyrus* *communis*	*PcoASR1*	2	1	202	22.05	5.7	−1.367	Nucleus
	*PcoASR2*	2	1	134	15.27	6.1	−1.436	Nucleus
	*PcoASR3*	2	1	135	15.46	6.48	−1.393	Nucleus
*Prunus mume*	*PmASR1*	1	1	148	15.18	5	−1.105	Nucleus
	*PmASR2*	2	1	141	15.94	6.19	−1.293	Nucleus
	*PmASR3*	3	2	223	23.85	5.66	−1.312	Nucleus
	*PmASR4*	4	3	234	26.03	5.66	−0.978	Nucleus
	*PmASR5*	2	1	66	7.66	6.2	−1.438	Nucleus
*Prunus persica*	*PryASR1*	2	1	98	11.03	6.4	−1.364	Nucleus
	*PryASR2*	2	1	193	20.76	5.68	−1.335	Nucleus
	*PryASR3*	2	1	311	33.22	5.62	−1.374	Nucleus
*Rubus occidentalis*	*RocASR1*	2	1	192	21.09	5.77	−1.319	Nucleus
*Malus × domestica*	*MdASR1*	2	1	202	22.13	5.74	−1.421	Nucleus
	*MdASR2*	2	1	133	15.08	6.41	−1.162	Nucleus
	*MdASR3*	2	1	120	13.22	5.48	−1.171	Nucleus
	*MdASR4*	2	1	182	19.8	6	−1.462	Nucleus
	*MdASR5*	2	1	200	22.27	5.91	−1.433	Nucleus
*Zea mays*	*ZmASR1*	2	1	138	15.54	5.89	−1.264	Nucleus
	*ZmASR2*	2	1	131	14.9	6.15	−1.373	Nucleus
	*ZmASR3*	2	1	269	27.78	5.07	−1.49	Nucleus
	*ZmASR4*	2	1	181	20.43	6.3	−1.28	Nucleus
	*ZmASR5*	2	1	104	11.79	6.65	−1.319	Nucleus, Cytoplasm
	*ZmASR6*	2	1	106	11.54	8.05	−1.125	Nucleus
	*ZmASR7*	1	1	106	12.08	10.56	−1.301	Nucleus
	*ZmASR8*	1	1	102	11.46	9.66	−1.19	Nucleus
	*ZmASR9*	1	1	102	11.41	9.78	−1.205	Nucleus
*Oryza sativa*	*OsASR1*	2	1	96	10.57	9.62	−1.222	Nucleus
	*OsASR2*	2	1	105	11.68	9.66	−1.212	Nucleus
	*OsASR3*	2	1	105	11.68	6.76	−1.112	Nucleus
	*OsASR4*	2	1	138	15.46	6.20	−1.243	Nucleus
	*OsASR5*	2	1	229	24.46	9.49	−1.313	Nucleus
	*OsASR6*	2	1	136	15.50	6.48	−1.113	Chloroplast, Cytoplasm, Nucleus

**Table 2 ijms-25-10283-t002:** The functions of *ASRs* in different plants in regulating fruit ripening process. References: [21,101,110,124,126,127].

Species	Gene Name	Description of Biological Functions
*Nicotiana tabacum*	*NtASR1*	Transgenic lines with reduced *ASR1* protein levels in tobacco show impaired glucose metabolism and altered levels of ABA and GA, which in turn regulate leaf Glc signaling and carbon partitioning.
*Prunus persica f.* *atropurpurea*	*PpASR1*	Transient expression of *PpASR* in tomato promotes fruit softening and ripening in cross-signaling between ABA and sucrose.
*Fragaria ananassa* *Duch.*	*FaASR1*	Overexpression of *ASR1* significantly increases the expression levels of anthocyanin biosynthesis and ABA biosynthesis genes in strawberry and further enhances the ripening and softening process of the strawberry fruit.
*Zea mays* L.	*ZmASR1*	The *ZmASR1* protein influences branched-chain amino acid biosynthesis and maintains kernel yield in maize under water-limited conditions.
*Solanum lycopersicum* L.	*SlASR1*	*SlASR1* enhances the accumulation of proline, methionine, valine, and isoleucine in tomato, thereby promoting fruit development.
*Oryza sativa* L.	*OsASR*	As a downstream component of the ABA signaling pathway, *OsASR* also increases the number of tillers and grain yield in rice via regulating ABA levels.

## Data Availability

Not applicable.

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
