# Peer review of "Development History, Structure, and Function of *ASR* (*Abscisic Acid-Stress-Ripening*) Transcription Factor"

_ijms, 2024, doi:10.3390/ijms251910283_

Round 1
Reviewer 1 Report
Comments and Suggestions for Authors
The review article provides a comprehensive overview of recent advancements in genes specific to abscisic acid, offering insights and a well-organized synthesis of the current state of research. The inclusion of diverse perspectives and thorough analysis of relevant studies significantly contributes to the understanding of the history, function, and structure of ASR genes.
However, there are areas where the review could be enhanced. Firstly, while the article provides a broad overview (growth, development, biotic and abiotic stress), it would benefit from a more detailed discussion. Incorporating more in-depth analysis or case studies in this area could provide readers with a clearer understanding of the implications. Secondly, the review occasionally lacks critical evaluation of the cited studies.
The author should address the following comments for further consideration:
1) Thoroughly revise the abstract. Summarize the content using at least 200 words to incorporate significant information.
2) Consider adding "transcription factor" to the title.
3) Add basic mechanisms and pathways in section 2.
4) Include phylogeny analysis of evolutionary aspects of these genes.
5) Clarify the criteria for selecting specific crops mentioned in table one.
6) Provide detailed gene structure, motif, and domain analysis to highlight specific differences.
7) Add a table of ASR genes identified in major crops related to fruit ripening.
8) Make sure all the crop-specific genes highlighted in Figure 1 are included in the text with proper references.
Author Response
Dear Editors and Reviewers:
We are very grateful to you for taking the time to read and modify our article again. We find that your comments play a very important role in improving the quality of our papers. We have carefully revised the paper in light of your comments, and please find our response to the comments made below. We marked the modified part of the manuscript in yellow.
Thank you for considering our revised manuscript!
1) Thoroughly revise the abstract. Summarize the content using at least 200 words to incorporate significant information.
Response 1:Thank you very much for your suggestion. We read the manuscript carefully and revise the abstract. For details please see the Abstract highlighted in yellow.
2) Consider adding "transcription factor" to the title.
Response 2: Thank you very much for your suggestion. We read the manuscript carefully and accepted your suggestions. We modified the title.
3) Add basic mechanisms and pathways in section 2.
Response 3: Thank you very much for your suggestion. We have carefully read the manuscript and added basic mechanisms and pathways in section 2. Please see the section 2 highlighted in yellow for details.
4) Include phylogeny analysis of evolutionary aspects of these genes.
Response 4: Thank you very much for your suggestion. We have carefully read the manuscript and added phylogeny analysis of evolutionary aspects of these genes. For details please see the Supplementary Figure S1.
5) Clarify the criteria for selecting specific crops mentioned in table one.
Response 5: Thank you very much for your question. The criteria for the selection of table one is based on species in which the function of each ASR gene has been preliminarily characterized in gene family analyses up to now.
6) Provide detailed gene structure, motif, and domain analysis to highlight specific differences.
Response 6: Thank you very much for your suggestion. We have carefully added detailed gene structure, motif, and domain analysis. For details please see the Supplementary Table S1 and Supplementary Figure S2 .
7) Add a table of ASR genes identified in major crops related to fruit ripening.
Response 7: Thank you very much for your suggestion. We have carefully added a table of ASR genes identified in major crops related to fruit ripening. For details please see the Table 2.
It was modified with yellow highlighting in line 264.
8) Make sure all the crop-specific genes highlighted in Figure 1 are included in the text with proper references.
Response 8: Thank you very much for your suggestion. We have carefully read the manuscript and modified part of the manuscript with yellow highlighting to ensure that all crop-specific genes highlighted in Figure 1. In addition, we have added all proper references.
Reviewer 2 Report
Comments and Suggestions for Authors
This article provides a good review of the topic, with an extensive and up-to-date bibliography analyzed and described in a way that is understandable to most interested readers. It covers the main topics, including abiotic stresses, growth, pathogens, fruit ripening, and also offers a thorough analysis of ASR genes for the major plant species.
However, I would like to highlight a few things: In the table, it seems that some cells have been left incomplete, especially in Oryza sativa. What is the reason for this? In this same table, I would like to find the genes of Arabidopsis thaliana. Such a widely used model plant and its ASR genes should appear in this review.
Author Response
Dear Editors and Reviewers:
We are very grateful to you for taking the time to read and modify our article again. We find that your comments play a very important role in improving the quality of our papers. We have carefully revised the paper in light of your comments, and please find our response to the comments made below. We marked the modified part of the manuscript in red.
1、This article provides a good review of the topic, with an extensive and up-to-date bibliography analyzed and described in a way that is understandable to most interested readers. It covers the main topics, including abiotic stresses, growth, pathogens, fruit ripening, and also offers a thorough analysis of ASR genes for the major plant species.
However, I would like to highlight a few things: In the table, it seems that some cells have been left incomplete, especially in Oryza sativa. What is the reason for this? In this same table, I would like to find the genes of Arabidopsis thaliana. Such a widely used model plant and its ASR genes should appear in this review.
Response 1:Thank you very much for your suggestion. We read the manuscript carefully and revise the table 1. We added complete information of Oryza sativa in the table. Please see table 1 in red for details. The first ASR gene, ASR1, was first identified in tomato (Solanum lycopersicum L.). So far, no homologous genes of ASR have been found in the model plant Arabidopsis thaliana [94].
Round 2
Reviewer 1 Report
Comments and Suggestions for Authors
It can bed accepted in present form for further consideration.